# Understanding the Dynamics of a Coastal Lagoon: Drivers, Exchanges, State of the Environment, Consequences and Responses

Samantha Chacón Abarca [1], Valeria Chávez [1], Rodolfo Silva [1], M. Luisa Martínez [2] and Giorgio Anfuso [3,*]

1   Instituto de Ingeniería, Universidad Nacional Autónoma de Mexico, Mexico City 04510, Mexico; SChaconA@iingen.unam.mx (S.C.A.); vchavezc@iingen.unam.mx (V.C.); rsilvac@iingen.unam.mx (R.S.)
2   Instituto de Ecología, A.C. (INECOL), Antigua Carretera a Coatepec no. 351, Xalapa, Veracruz 91073, Mexico; marisa.martinez@inecol.mx
3   Faculty of Marine and Environmental Sciences, University of Cádiz, 11071 Cádiz, Spain
*   Correspondence: giorgio.anfuso@uca.es

**Abstract:** At present, many coastal ecosystems worldwide are highly affected by anthropic activities. La Mancha lagoon, in the state of Veracruz, Mexico, is an important ecosystem due to the wide array of ecosystem services that it provides. In this paper, an analysis of the environmental balances of the lagoon is outlined, using the Drivers, Exchanges, State of the Environment, Consequences and Responses (DESCR) tool. The methodological framework considers the interrelationships between the natural systems and the forces of change that alter the performance of the natural environment, in order to provide an overview of actions that may reduce negative consequences. The study area has been impacted by anthropic development, such as changes in land use for agricultural and livestock activities, loss of mangroves due to logging and modifications, carried out by local fishermen, to the natural hydrodynamics of the lagoon that alter the salinity and affect the ecosystem dynamics. Following analysis of the area, using the DESCR tool, the responses proposed include long-term environmental impact evaluation, with the aim of preserving the local coastal ecosystems.

**Keywords:** La Mancha lagoon; coastal; dynamics; DESCR framework

## 1. Introduction

Coastal zones are amongst the most dynamic areas of the planet, where processes occurring in the atmosphere, hydrosphere, lithosphere and biosphere are endlessly interacting in a continually changing balance. For human beings, these balances are highly relevant, as the coasts are home to socio-cultural and economic processes vitally important to humankind [1]. Benefits provided by these systems drive their exploitation, often leading to severe degradation [2]. Such negative changes stimulated the need for coastal study and monitoring in recent years [3].

The reduction in space available for the natural functioning of coastal ecosystems, due to phenomena such as sea level rise and the construction of infrastructure, produces what is known as "coastal squeeze", which may lead to the disappearance of some species, ecosystems and, consequently, ecosystem services. According to Silva et al. [4], this phenomenon includes local, regional or global anthropogenic processes and favors negative consequences at different time scales that induce alterations in the natural dynamic of ecosystems and inhibit the capacity of ecosystems to adapt to climate change. Human actions are therefore one of the most important pressures faced by ecosystems [1].

The Drivers–Pressure–State–Impact–Response (DPSIR) framework [5] has been widely used to assess coastal squeeze. However, in 2016, Elliot et al. [6] reported that, since 1999, 25 schemes for management and decision making across ecosystems have used derivations of the DPSIR conceptual framework. Elliot et al. [6] recognized that clearer,

more comprehensive, nested conceptual models are needed to quantify the links between pressure–state change in marine and coastal ecosystems.

The Drivers, Exchanges, State of the Environment, Consequences and Responses (DESCR) framework is based on the Drivers–Pressure–State–Impact–Response (DPSIR) framework, which is considered a useful system for the organization and presentation of environmental sustainability factors [5]. DESCR is a variant of this methodology, in which, instead of analyzing the pressures, the bidirectional exchanges of fluxes of matter and energy with the environment are considered simultaneously, evaluating their natural dynamics and connectivity. These exchanges, in which the drivers modify the state of the environment and vice versa, are considered in this cycle, as suggested by [7]. This analysis of the exchanges reflects the intensity of the pressures that exist in the study area. The impacts are evaluated as consequences in the ecosystems and can be positive, negative or neutral [4]. This methodology was described in a study on coastal squeeze in Puerto Morelos, Mexico, as a tool to evaluate and manage the consequences of the phenomenon [4].

Over recent years, environmental management has developed considerably in Latin America, but problems associated with anthropic factors such as pollution, destruction and degradation of renewable natural resources and the environment are still all too common [8]. In the case of Mexico, the tourist boom of recent years [9] has left its mark. In the state of Veracruz, the coastline, ca. 745 km in length [10], is dotted with urban and touristic developments and is of great importance ecologically, socially and economically [9]. In particular, the coastal lagoon of La Mancha (Figure 1), in the state of Veracruz, belongs to the municipality of Actopan. The lagoon has been on the Ramsar List of Wetlands of International Importance since 2004 [11] and was also established as a Priority Mangrove Conservation Site by CONABIO, the National Commission for the Knowledge and Use of Biodiversity [12]. Tourism in the vicinity of La Mancha is managed by a sustainable ecotourism company, called *La Mancha en Movimiento S.S.S.*, where local fishermen, housewives, farmers and others have been trained to work as eco-guides [13].

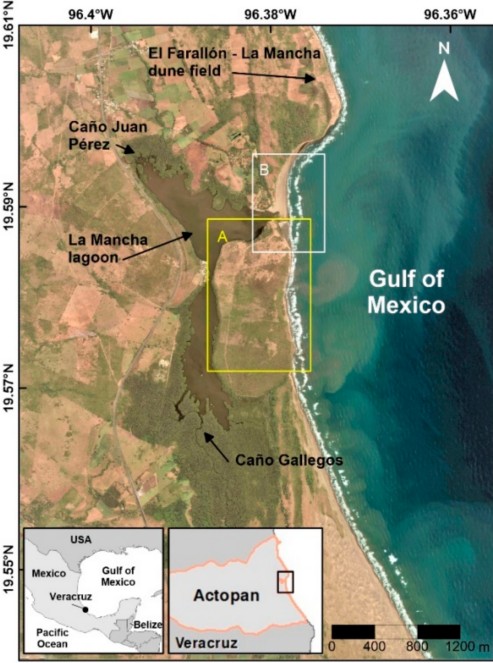

**Figure 1.** Location of the study area (satellite image source [14]). Area A is the site of a recent urban development project. Area B is the beach and lagoon inlet area.

It is important to identify, describe and analyze the anthropic and natural actions that regulate the lagoon ecosystem of La Mancha, in order to understand the conservation status of this environment and subsequently generate viable responses to manage it in a

sustainable way. To explain the development and evolution of the coastal ecosystems of the area around La Mancha, a literature review was carried out, analyzing physical and environmental characteristics of the area, impacts of anthropic activities and consequences of the pressures on the lagoon, dune and mangrove ecosystems in the area, according to the DESCR framework. The characteristics of the lagoon and associated ecosystems were analyzed from a qualitative perspective. This approach could be used as a basis to outline policies for the sound management and use of this area.

## 2. Materials and Methods

### 2.1. Study Area

The study site is on the Gulf of Mexico in the state of Veracruz, between the coordinates 19°33′ and 19°36′ N and 96°22′ and 96°24′ W (Figure 1). Geologically, this area is a cumulative plain, formed by lacustrine, fluvial and biogenic sediments, occasionally combined with marine deposits [12]. The geomorphological evolution of the site is marked by long-term minimal marine sediment input and intense aeolian transport [15], which are both recorded during storm conditions related to northerly winds. Up until the 1970s, the aeolian transport had favored the formation of significant coastal dunes and active and fully mobile transverse dune fields. From the 1980s onwards, these dunes, particularly extensive on the north side of the rocky ledges (Figure 1), have stabilized naturally [16].

The lagoon of La Mancha is a brackish coastal lagoon, liable to siltation, with a surface of 126 ha, an average depth of 1.4 m and a maximum depth of 3 m [12,17]. The tides are mixed diurnal with a range of 0.69 m [18], and the lagoon is fed by two permanent river tributaries in the dry season (see Figure 1) and intermittently communicates with the sea [18]. The balance among sea water, rain and groundwater discharges regulates the hydrological variability of the lagoon [12]. The beach communicates with the lagoon in the lagoon inlet, and its natural opening and closing lead to modifications in the environment that alter the dynamics of the lagoon and cause changes in other surrounding ecosystems. The hydrodynamics of the inlet are very dynamic; in winter, a sandbar is formed, which naturally disappears again during the rainy season [9] (see Figure 2). The opening and closing of the lagoon inlet determine the migration of sediments, fish species and nutrients into the lagoon body [18]: when the inlet of the lagoon is closed, the water level of the lagoon rises and the salinity decreases; when it is open, the water level decreases and there is greater salinity and sediment accumulation [15] (see Figure 2). The lagoon inlet is also opened by local fishermen about three times per year in the dry season, in order to increase their catch by altering the normal flooding regime [12].

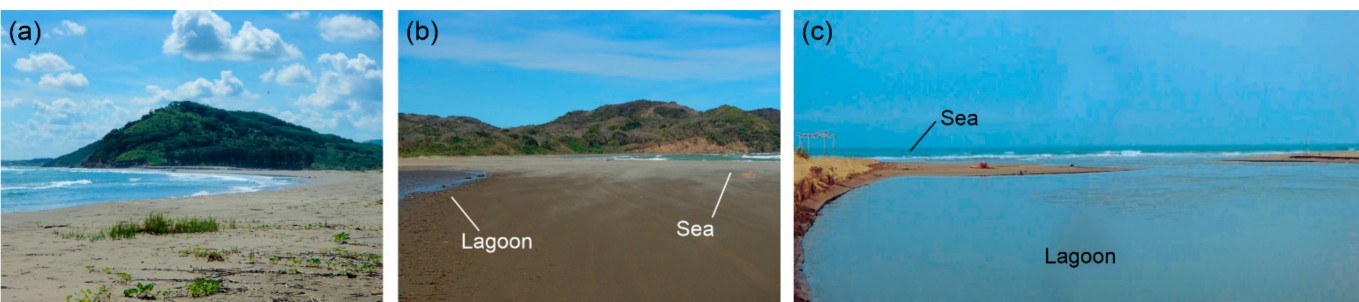

**Figure 2.** La Mancha: (**a**) beach; (**b**) the lagoon inlet closed by the sandbar in the dry season; (**c**) the open lagoon inlet in the wet season.

The climate of the study area has three pronounced seasons: the rainy season, from June to October; a period of "Nortes", winter storms that occur in the western Gulf of Mexico, from November to March; and a dry season, in April and May [12].

There is a wide variety of natural ecosystems in the study area, including mangroves around the lagoon (Figure 3a,b), coastal dune vegetation (Figure 3c), rainforest, tropical grassland, swamps, secondary vegetation, crop pasture and fruit trees. Mobile dune fields

were reported to be mostly vegetated with *C. punctatus* and *Palafoxia lindenii* in 2008 [16], but recently, *Schizachyrium scoparium* became dominant and replaced those species (Figure 3d).

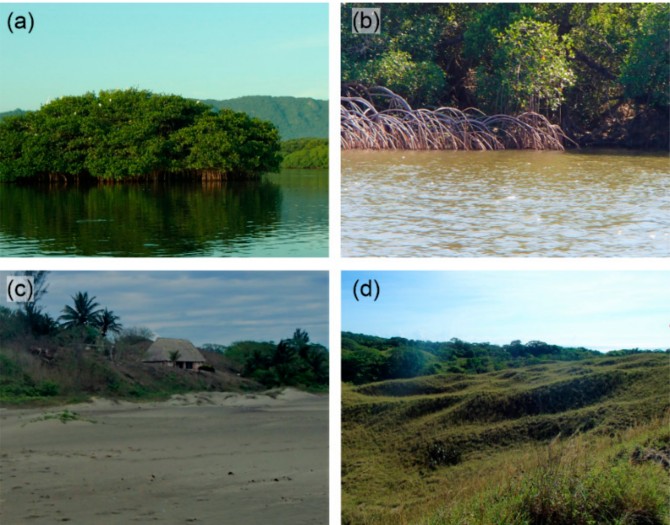

**Figure 3.** La Mancha: red mangrove in the (**a**) lagoon and (**b**) lagoon margins (*Rhizophora mangle*); (**c**) beach and coastal dunes; (**d**) vegetated dune field.

As the study area forms part of one of the world's largest migratory corridors for birds of prey, La Mancha receives many visiting birds of prey, shorebirds and waterfowl species, as well as marine and terrestrial fauna [11]. The coastal wetlands of the study area provide ecosystem services, such as water filtration, temperature regulation and storm protection, and goods such as fish, as well [19].

Other ecosystem services provided are shelter for animal species in their youth, scenic value for ecotourism and serving as a carbon dioxide sink [12]. The productivity of the mangrove ecosystem can be measured in terms of its primary components, i.e., litter production [20], which is a source of nutrients for the species inhabiting this ecosystem. Utrera-López et al. [20] stated that the mangrove in the area of La Mancha has medium-high litter production, with annual values of 6.92–13.50 t/ha/year.

*2.2. DESCR Framework*

The review of available information was conducted in December 2020, by searching the Scopus, Google Scholar and the *Instituto de Ecología, A. C.* databases, using the following keywords: lagoon, dunes, mangroves, coastal erosion and land use, in association with La Mancha, Veracruz and Mexico. Additionally, based on the authors' knowledge, the names of specific authors who have worked on coastal ecosystems and coastal evolution in this area were searched: M.L. Martínez, P. Moreno Casasola and J. López-Portillo. Masters and PhD theses were also consulted, as well as the gray literature and maritime climate databases. A list of relevant references is presented in Appendix A.

The information thus obtained was analyzed in line with the DESCR methodology [4], consisting of the following:

1. Drivers, as the social, economic or environmental forces that put pressure on the environment, which are both natural (storms, hurricanes, etc.) and human (urbanization, pollution, etc.);
2. Exchanges, which are the measure of how much the driving forces have produced changes in the system;
3. The State of the Environment, where the current environmental conditions are defined after analyzing the pressures;

4.   Consequences, which are the effects of the processes on the environment, e.g., sediment transport may lead to erosion/accretion processes, and loss of ecosystem services;
5.   Responses, which are the mitigation actions adopted by local authorities and stakeholders to solve environmental problems and/or improve the quality of the environment.

## 3. Results

### 3.1. Drivers in the DESCR Framework

3.1.1. Natural Drivers

Coastal ecosystems are affected by long- and short-term fluctuations in sea states and spatial changes on the coast [21], induced by the action of currents, winds and waves produced by natural phenomena such as hurricanes and storms [22]. The location of La Mancha, in the southwest of the Gulf of Mexico (Figure 1), means its dynamics are mainly the result of phenomena occurring within the gulf. The main natural drivers in La Mancha are as follows:

1.   Winds;
2.   Waves;
3.   Storms, tropical cyclones and hurricanes.

Winds

Figure 4 shows the annual and seasonal wind roses for the coordinates 19.5° N 96° W, the location of which is approximately 40 km seaward of the La Mancha lagoon inlet. The data were obtained from the Era5 Climate Reanalysis produced by the European Centre for Medium-Range Weather Forecasts (ECMWF): hourly data on sea state parameters for 1979–2019, on a regular latitude–longitude grid at 0.5° × 0.5° resolution [23]. The annual Climate Reanalysis shows that the most frequent winds in La Mancha come from the NW, with speeds of 20–35 km/h. During winter storms and in the rainy season, winds reaching >50 km/h are observed.

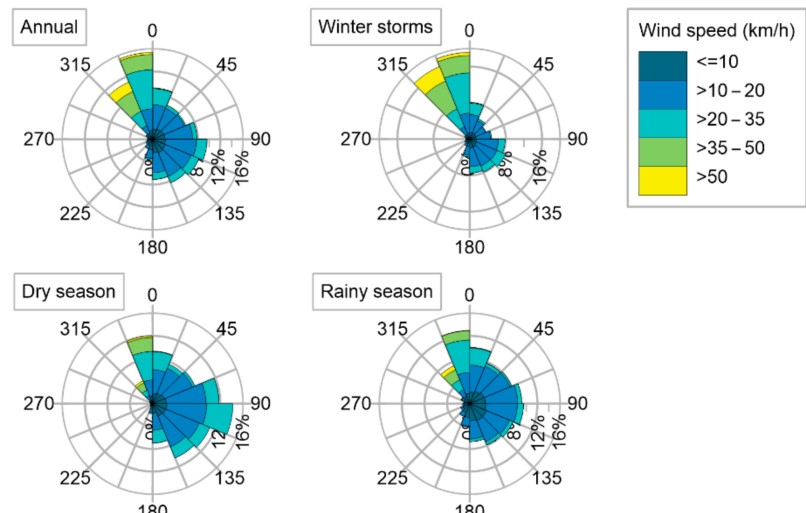

**Figure 4.** Annual and seasonal wind roses for La Mancha, data from Era5 [23].

Waves

Wave roses (Figure 5) for the 19.5° N 96° W coastal cell, obtained from the Era5 database [23], show that the most energetic waves come from the NE, with a 30% frequency and heights of 0.5–1 m. In winter, waves approach from the NE, with heights of 1–1.5 m, and the highest wave heights come from the NW (315° to 360°). Overall, the highest wave heights occur during winter storms, and the lowest values are in the dry and rainy seasons.

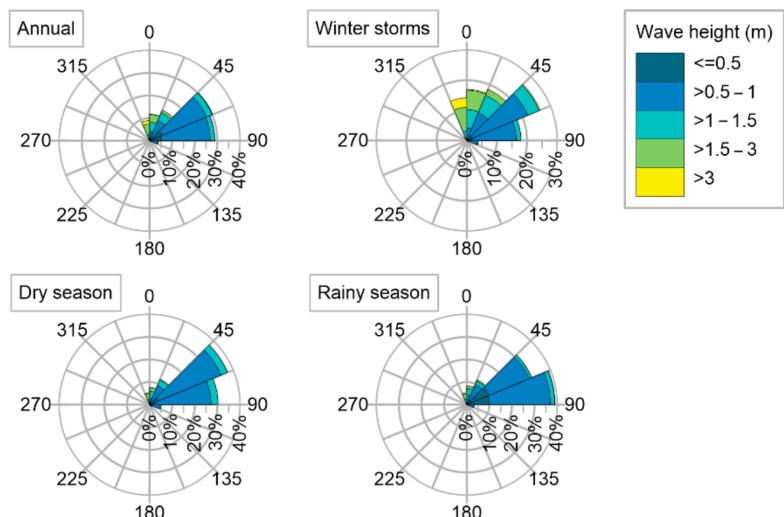

**Figure 5.** Annual and seasonal wave roses for La Mancha, data from Era5 [23].

Storms, Tropical Cyclones and Hurricanes

According to Martínez et al. [24], Veracruz has a sui generis climatic confluence, caused by winds and atmospheric disturbances, which can generate considerable flooding on the coast, since the rivers overflow, and the phreatic level rises. The prevailing winds in Veracruz are generated by high-pressure systems in the winter and low-pressure systems in the summer. These atmospheric systems usually approach from the north and the east, impacting the coast of Veracruz [25], with winds of up to 100 km/h in the summer, and up to 180 km/h in the winter [24]. In the last 50 years, six hurricanes of category 3–5 have hit the area [24]. Although hurricanes are not very frequent, the area is at risk from flooding and erosion due to the energetic waves and storm surge induced by these phenomena [24].

Figure 6 shows the annual wind and wave roses for extreme events for the area of La Mancha. The data correspond to the coordinates 19.50° N 96.25° W and were obtained from the Maritime Climate Atlas of *Instituto de Ingeniería UNAM* [26], calculated from 1948 to 2010 with the WAM-HURAC numerical model [27]. The wind rose includes hourly data with a wind velocity of >50 km/h (0.29% of the 1948–2010 time series). The wave rose includes hourly data with a wave height of >5 m (0.17% of the 1948–2010 time series). It can be seen that the more energetic winds and waves come from the north.

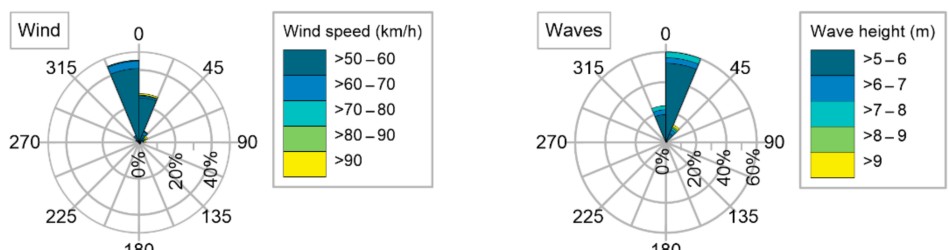

**Figure 6.** Annual wind and wave roses for extreme events for La Mancha, data from [26].

Tides

The measured tidal levels near the study area are reported by the Secretaría de Marina (SEMAR, Mexican Secretariat of the Navy) for two different locations: Tuxpan (97°20′48″ W 20°57′12″ N) [28], and Veracruz (96°07′51″ W 19°12′03″ N) [29]. These values are estimated with measured data from July 1999 to December 2017 and are presented in Table 1. The values for La Mancha in Table 1 were calculated with a linear interpolation between the data of Tuxpan and Veracruz.

**Table 1.** Tidal levels referring to the MLLW (mean lower low water), data from SEMAR [28,29].

| Tidal Level | Tuxpan | La Mancha [1] | Veracruz |
|---|---|---|---|
| HAT (Highest Astronomical Tide) (m) | 1.040 | 1.087 | 1.100 |
| MHHW (Mean Higher High Water) (m) | 0.496 | 0.500 | 0.501 |
| MHW (Mean High Water) (m) | 0.410 | 0.415 | 0.417 |
| MSL (Mean Sea Level) (m) | 0.284 | 0.286 | 0.287 |
| MLW (Mean Low Water) (m) | 0.220 | 0.230 | 0.233 |
| MLLW (Mean Lower Low Water) (m) | 0.000 | 0.000 | 0.000 |
| LAT (Lowest Astronomical Tide) (m) | −0.470 | −0.493 | −0.500 |

[1] Interpolated values.

3.1.2. Anthropic Drivers

The basin of La Mancha is a rural area with little urban development, where incomes are based on livestock and agricultural activities. The contamination due to untreated wastewater from nearby settlements, and the small-scale felling of mangrove trees for agricultural land are the main problems in the study area, regarding the health and integrity of the ecosystems [18]. Notwithstanding, Ramírez Méndez [18] underlined the impacts of the opening of the lagoon inlet, two or three times a year, which modifies the natural hydrosedimentary regime in the lagoon. These openings are carried out by the *Sociedad Cooperativa de Producción Pesquera La Mancha de S.C.L.* (Fisheries Production Cooperative of La Mancha), in order to improve their fish catches [12]. Most of these fishermen prefer to fish inside the lagoon (72%), and almost half of them (42%) fish all year round. Consequently, they deliberately alter their fishing grounds [18].

The anthropic drivers considered are as follows:

1. Tourism;
2. Planed urban infrastructure.

Tourism in Coastal Areas and the Effects of Dune Trampling

The *Centro de Investigaciones costeras "La Mancha"* (CICOLMA) is a research center which aims to generate knowledge related to local ecological processes. This knowledge is transferred through activities such as ecotourism programs. In this case, people in the local communities who are interested in the conservation of the area are trained as eco-guides, and a sustainable business model has been developed, which includes ecotourism, training activities and fishing [30].

Hesp et al. [31] stated that in the dunes of La Mancha, pressure from tourists (see Figure 1) causes trampling, which can become a relevant disturbance factor. They pointed out that natural dunes are among the most sensitive environments to trampling. They also suggested that there is a relationship between the slope of the footpath and the disturbance to the dune vegetation and considered the height of the dune an important factor when quantifying the trampling damage to these ecosystems through trampling. As the dunes are trampled on, sediment dynamics may be altered, and this could indirectly affect the lagoon. Due to the feedback between dunes and the beach, modifications of dune dynamics may affect the sand budget of the beach which can eventually alter the opening and closure of the inlet. Psuty et al. [15] described the complex, indirect dynamics of the study site.

Planned Urban Infrastructure

A real estate project was begun in 2018 by *Innova Dintel Guanajuato*, with 800 "sustainable" houses planned for a 167 ha site (Figure 7), as well as a bridge, or connecting dock [32]. The land and houses in this area have been advertised for sale in 2021, and access roads to this area have also been laid, modifying the natural environment (Figure 7). Local people are concerned by the invasive nature of this project, citing the massive increase in the urban density and the environmental impacts produced by the construction work [33]. Once built, it is feared that these "sustainable" houses will use water extracted from the lagoon, which would cause an imbalance in the system. Furthermore, regulations are

needed to ensure that wastewater is not discharged into the sea or the lagoon without proper treatment.

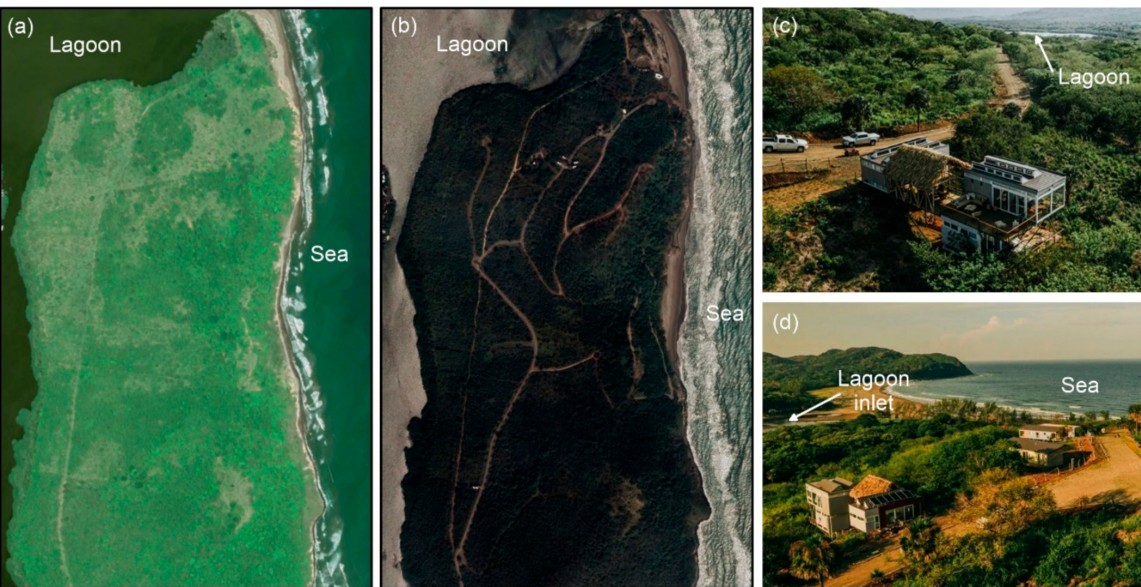

**Figure 7.** The area of the urban project in La Mancha (Area A in Figure 1): (**a**) in 2017, before construction (adapted from Google Earth); (**b**) in 2021, with new roads (adapted from Google Earth); (**c**,**d**) some of the new houses in the area (adapted from [34,35], respectively).

### 3.2. Exhanges in the DESCR Framework

The exchanges and the characteristics of the drivers will eventually be affected by the changes in the state of the environment [4]. The main exchanges in the study area are shown in Figure 8.

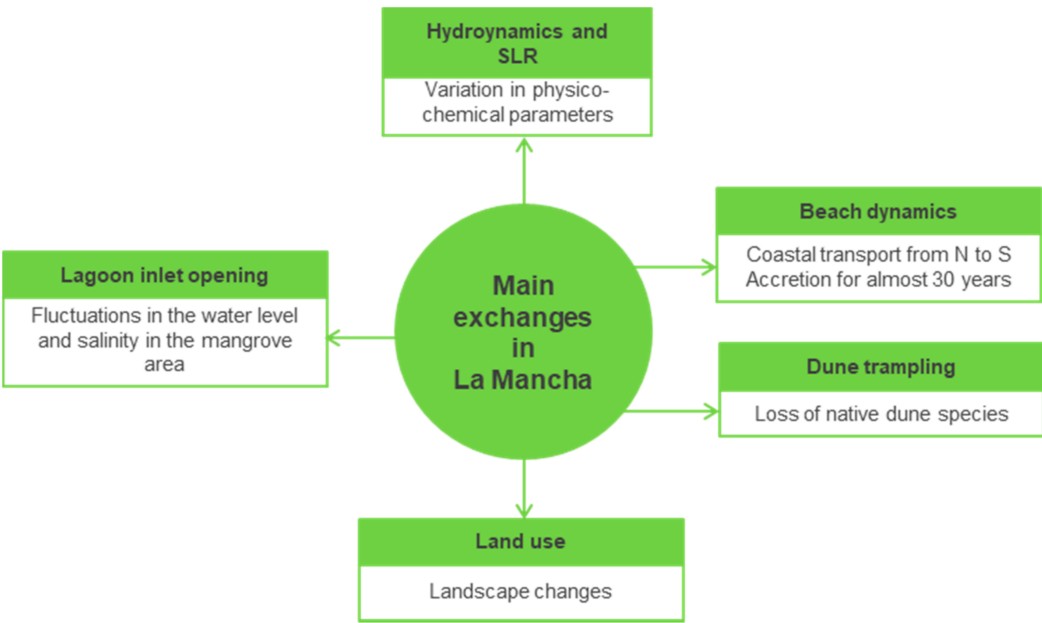

**Figure 8.** Main physical exchanges in La Mancha.

### 3.2.1. Hydrodynamics and SLR

In August (the rainy season), the values of pH, salinity and density are affected by the amount of freshwater available. In April (the dry season), these parameters depend only

on the sea water, and this is also when the greatest spatial variability of the temperature is observed.

Under normal conditions, the salinity in the lagoon is stratified from south to north: the Caño Gallegos stream provides freshwater, with values of less than 0.5 PSU, while in the north of the lagoon, the salinity is higher, since it is connected to the sea [12]. As it was mentioned previously, the opening of the lagoon inlet by local fishermen also affects the salinity of the lagoon. From the results of the numerical model implemented by Rivera [36], it is seen that if the inlet is open continuously for more than three months, the residence time of the water in the lagoon increases. This increase is directly related to eutrophication in coastal waters [37].

Regarding coastal flooding, storm surge levels near the study area were reported by Silva et al. [38], where the values of the storm surge caused by the atmospheric pressure gradient, wind and waves were calculated for different return periods using climate data from 1948 to 2010. These values are shown in Table 2.

**Table 2.** Storm surge levels in La Mancha area, data from Silva et al. [38].

| Storm Surge | Return Period (Years) | | | | | | | | | |
|---|---|---|---|---|---|---|---|---|---|---|
| | 2 | 5 | 10 | 15 | 20 | 25 | 30 | 40 | 50 | 100 |
| Atmospheric pressure gradient storm surge (m) | 0.03 | 0.11 | 0.17 | 0.20 | 0.22 | 0.23 | 0.24 | 0.26 | 0.28 | 0.32 |
| Wind storm surge (m) | 0.06 | 0.16 | 0.26 | 0.33 | 0.38 | 0.42 | 0.45 | 0.50 | 0.54 | 0.68 |
| Wave storm surge (m) | 1.27 | 1.40 | 1.47 | 1.51 | 1.54 | 1.58 | 1.62 | 1.66 | 1.70 | 1.80 |
| Pressure + wind + wave storm surge (m) | 1.36 | 1.67 | 1.90 | 2.04 | 2.14 | 2.23 | 2.31 | 2.42 | 2.52 | 2.80 |

The relative rise in the sea level has a range of effects on the morphological changes in this area at different scales [19]. Sea level rise affects the flood levels, which are related to the geographic location, coastal orientation and beach slope [24]. According to Pérez et al. [19], in Veracruz, the trend of sea level rise is +1.89 mm/year. This is in line with global IPCC reports and is conditioned by the subsidence of the natural deltaic surface [39]. The geodynamics of the Gulf of Mexico are regulated by the movements of the North America, Circum-Pacific Kula, Farallón, Cocos and Caribbean tectonic plates. These tectonic plates cause continental and oceanic movements that intermittently induce the continental accumulations of deep marine sediments in the continental marginal accretional prism of the Gulf of Mexico [40].

3.2.2. Problems Affecting the Vegetation and Fauna

The structure and composition of the mangroves depend on a variety of factors, i.e., oceanographic, climatic, geomorphological, edaphic conditions, level and duration of flooding and sediment load [41]. According to several studies, the location of the mangrove forest affects the diameter and height of some specimens: they grow the most in areas protected from wave action. Moreno-Casasola et al. [42] also listed other variables relevant for mangroves in freshwater wetlands: salinity, water regime, conductivity and redox potential.

The dynamics of the vegetation around La Mancha are also clearly related to the opening of the inlet. When the inlet is closed and the water depth of the lagoon reaches <120 cm near the sandbar, the inlet opens naturally, discharging the supratidal accumulation through it. This ebb and flood exchange of water causes variations in parameters such as salinity and sediment distribution. The inlet dynamics are essential for the mangroves: if the inlet is permanently closed, there will be excessive flooding, and the mangroves will die due to a lack of oxygen in the stagnant water; if the inlet is permanently open, salinity will increase, also causing mortality [15].

Animal species are also affected by fluctuations in the water level and the salinity of the lagoon. For instance, oysters, and some other species, have high mortality rates when the water level increases and the oxygen in the water decreases, leading the fishermen to

open the inlet, which causes imbalances in the lagoon dynamics [15]. Other environmental factors, such as evaporation and insolation, are <1% significant for mangrove development here [43].

Regarding the vegetation that develops on the coastal dunes surrounding the lagoon, there has been a decrease in the species which were previously most abundant. Previous studies indicated that although trampling on the dunes is infrequent, it is a relevant factor in the area, resulting in vegetation loss and compaction of the sand [31]. The seedlings are totally destroyed by crushing and flattening, leaving only bare soil in some cases [31]. The intensity of trampling is best seen on the steeper slopes, where the vegetation is detached and crushed [44]. The occurrence of this activity is not widespread, and trampling has a low impact; however, it may induce local extinction of some plant species [31].

*3.3. State of the Environment in the DESCR Framework*

Cortés [44] described the beach at La Mancha as having a dissipative profile with a bar, following the classification of Masselink and Short [45]. The sedimentary input comes from the northern dune field, with a net north-to-south longshore sediment transport. Ramírez Méndez [18] identified an overall trend of beach accretion, recording an advance of 58 m for 2005–2015, i.e., 0.109 km$^2$ of beach. Chávez [46] reported that the beach sediment is a fine sand (according to the ASTM D 2487 norm), with a mean diameter of 0.176–0.305 mm.

The main geomorphological components at the inlet are as follows: the tidal flood delta, and the sedimentary bar on the northern side of the inlet, which is a storm berm and is part of the beach (see Figure 9, the area corresponds to Area B in Figure 1). The berm is built up by sediment during storm surge conditions (winter storms and hurricanes). When there is a sediment deficiency, usually associated with erosive wave conditions, the berm breaks up [15]. The inlet is usually open from August to November and closed in December, with no definite pattern from January to September [18]. When the sedimentary bar is low and the inlet is open for a long period, intertidal exchanges sequester sediments in the north of the lagoon, the area of the tidal flood delta extends and this part of the lagoon becomes shallower [15].

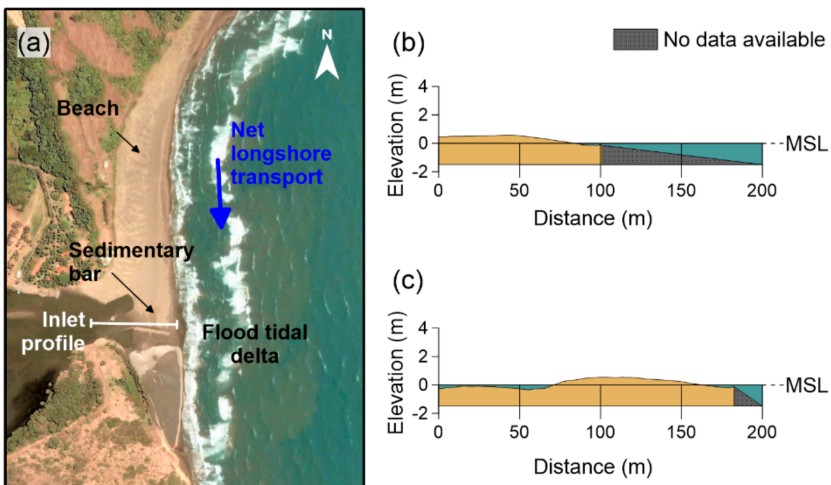

**Figure 9.** (**a**) Inlet area, and inlet profiles in La Mancha measured during winter storms in November (**b**) 2013 and (**c**) 2014 (data from [12]), see Area B in Figure 1.

3.3.1. Lagoon Vegetation

The physiognomy of the plant communities which surround the lagoon can be divided into tree-dominated and herbaceous species. Around the lagoon (Figure 1), there is a heterogeneous mangrove forest of about 190 ha, with a height range of 5–15 m and typical, associated plants [18]. Mangrove species found here include *Rhizophora mangle* (red mangrove), *Avicennia germinans* (black mangrove), *Laguncularia racemosa* (white mangrove)

and *Conocarpus erectus* (button mangrove), with *Avicennia* and *Laguncularia* being the most abundant (88%) [15] (Figure 10). Between the 1980s and 2010, the area of the mangrove fell by 33.8%, although another 0.3% of mangrove area was converted from agricultural livestock to mangrove, between 2005 and 2010 [18].

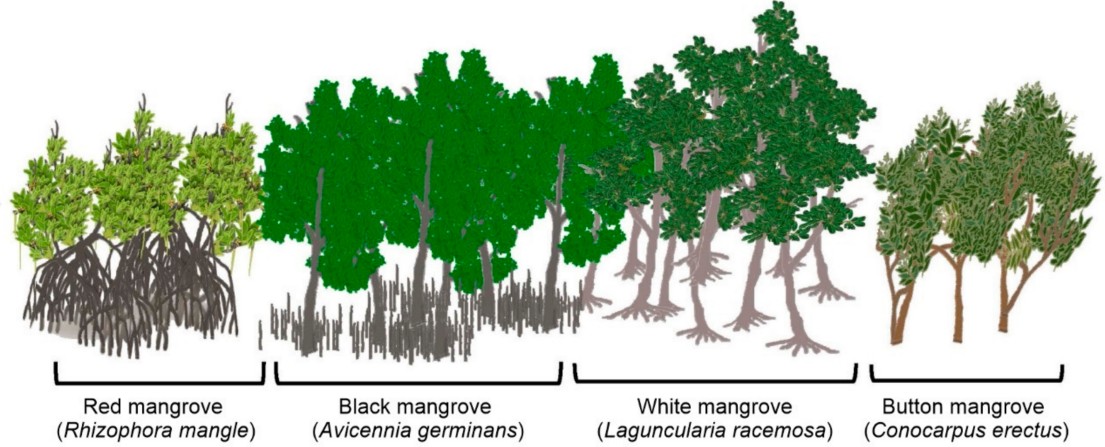

**Figure 10.** Distribution of mangrove species in La Mancha.

The hydrological and geomorphological conditions of the area determine the development of the species there [17], which grow the most in periods of flooding. The authors of [17] stated that the tallest mangroves are found in the area affected by floods, since this is where there are most nutrients. As for the health of the mangroves, this depends on the season, where more robust trees are seen in the rainy season.

Psuty et al. [15] explained the importance of flooding, as the seedlings of some species, such as *Avicennia,* are dispersed by water currents at the end of the rainy season, germinating when the sandbank closes the inlet, and the water level starts to rise. Litter production is an important factor to be taken into account in the evaluation of mangrove ecosystems, and the amount of fallen leaves in this ecosystem in La Mancha is 1025 g/m$^2$ per year [17].

### 3.3.2. Anthropic Impacts

While the state of Veracruz is very densely populated, that is not the case of the study area [44]. The land use and vegetation for 2015 [47] are shown in Figure 11 at a 1:250,000 scale. The main land uses are agriculture and pasture. According to Ramírez Méndez [18], the natural landscape of La Mancha is fragmented due to anthropogenic activities related to agriculture, livestock, fishing, tourism and human development, which have mainly affected the low deciduous forest.

### *3.4. Consequences in the DESCR Framework*

### 3.4.1. Beach and Dunes

According to Doody [48], a beach that is extending offshore due to accretion needs sediment. As the amount of sediment increases and the sediment balance of the beach moves from neutral to positive, large dune areas will develop, causing regenerative trends in the terrestrial ecosystems. In the study area, the coastline and the vegetation bordering it have been altered. The aerial photographs in the analysis of Psuty et al. [15], in the 1980s, show mobile coastal dunes moving across the headland, supplying the beach with sand. However, by the mid-1990s, vegetation had covered these mobile dunes. This trend has continued, and, with less sand available, sediment transport to the south has decreased. In addition, as the use of 90 ha of the mobile sand dunes has changed, only 10% of the original area of mobile dunes is now in its previous, natural state.

Regarding the morphology of the inlet, there is a southward sediment transport. As sediment is now scarce, the berm is less developed, which means that the inlet stays open for longer. This causes the tidal exchanges to sequester sediment in the northern part of the

lagoon, increasing the area of the delta, inducing subsidence and deepening the northern part of the lagoon [15].

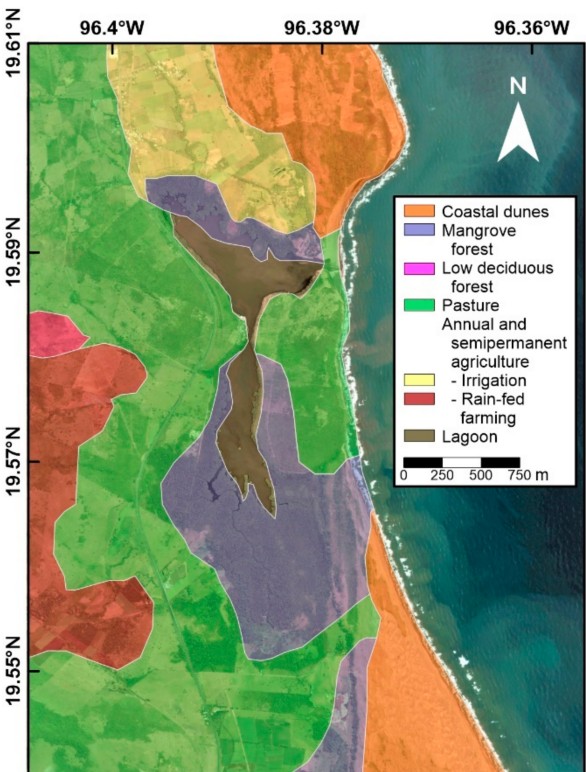

**Figure 11.** Land use and vegetation in La Mancha. Data from INEGI [47].

3.4.2. Lagoon and Mangroves

According to Psuty et al. [15], the variability of the water level and salinity has important effects on the species growing in the mangrove ecosystem. The opening and closing of the inlet are the determining factors in the behavior of this system; the closure of the bar, which prevents entry of seawater to the lagoon, produces a decrease in oxygen, salinity, pH and chlorophyll [18]. Some fish species in the lagoon change their balances depending on whether the inlet is open or closed, and the existence, or dominance, of many species varies depending on the quality of the lagoon water [15].

The mangrove species in the area provide shelter to native and endemic species, such as tilcampo, crocodiles, iguanas and turtles, which are vulnerable to ecosystem loss; crabs, ocelots and tigrillos, which are in danger of extinction; and also flounder, bass, yellowfin crappie and striped crappie, which are valuable commercial species for the local fishermen [18]. Mangrove is also an ecosystem that acts as an ecological membrane. Martínez et al. [49] analyzed the accumulation of heavy metals in La Mancha, noting that it was greater in mangrove leaves than in nearby sediments, noting the ability of this ecosystem to protect and regulate the balance and mitigate the effects of these substances in the environment, thus decreasing marine pollution.

Psuty et al. [15] pointed out that the germination of seedlings may eventually become more limited, depending on the opening and closing of the inlet, as some species, such as *Avicennia* seedlings, are dispersed by the water currents at the end of the rainy season (September–October), and they germinate when the sandbar closes the inlet.

*3.5. Responses in the DESCR Framework*

As the dynamic equilibrium of the lagoon system depends on changes in the physico-chemical parameters, the artificial opening of the inlet alters the natural dynamics. Since the opening of the inlet is beneficial to the fishermen's activities, the magnitude of the

long-term ecological consequences of this practice should be quantified, taking into consideration large-scale processes such as SLR, in order to devise an adequate management plan.

Human activity has played an important role in the destabilization of the beach and dunes by modifying the hydrodynamics, the sediment availability and the sediment transport patterns. It is therefore essential to create conditions for a sand reservoir, and alternatives to preserve the vegetation of this ecosystem [48]. Trampling and other activities, such as the use of scooters, put the dune vegetation and their natural dynamics at risk. However, tourism benefits many people in the area. Therefore, ecotourism plans could be focused on the dunes, where the cultural ecosystem services can be taken advantage of while seeking the sustainability of the ecosystem. To restore the coastal dunes, it is necessary to restore the natural dynamics and recover habitats for dune-building species (e.g., previously, the mobile dunes were mostly covered by the grass *Schizachyrium scoparium*). Native species such as *Croton punctatus*, *Palafoxia lindenii* and *Chamaecrista chamaecristoides* are abundant on mobile dunes, and thus actions are needed to re-establish their environment so that they can increase their plant cover. Fencing the dunes, in order to restrict access to them, to reduce air flow and to increase sand deposition, is also an option to restore the equilibrium [50].

Regarding the recent urban development, this may have intense repercussions on the environment, as well as on the socio-economic development of the population of the area. Despite the fact that Mexico has legal instruments to conserve and protect natural ecosystems (General Law of Ecological Balance and Environmental Protection, General Law of Wildlife, General Law of Sustainable Forest Development, Fisheries Law), there is a lack of commitment to promoting sustainable development objectives by the state. The conflict is part of the conservation versus extraction debate. The "Comprehensive Management Plan for the La Mancha-El Llano Basin" (2006) was conceived as an instrument that links communities, academia and government authorities and allows discussion of the current environmental situation and conflicts, evaluating the participation of the actors in the area [51]. This document focuses on: (i) the biological and cultural conservation and restoration of the ecosystems and landscapes of the basin, (ii) community participation and (iii) comprehensive planning, taking into account the management of the area [52]. Despite the implementation of this plan, the resistance to change of the population is both noticeable and very intense, creating conflicts with government entities and within the population in general, based on economic interests [52]. This is currently occurring with the development of the "Diada" project in Veracruz which, according to Gómez Ramirez [53], will probably favor only wealthy people, and the local, native population will see hardly any economic benefits.

It is necessary to carry out an environmental impact assessment of the anthropogenic expansion in the area, since this can cause fragmentation of ecosystems and put some species at risk, such as migratory birds [53]. Mendoza et al. [54] analyzed the changes in land use and the valuation of ecosystem services on the coast of the Gulf of Mexico, noting the importance of maintaining a healthy balance in the area, related to the value and provision that this system provides to the community. It may be useful to carry out an assessment of ecosystem services in the area and determine how the urbanization projects will alter them, eventually affecting society. Environmental regulations should look for alternatives that promote conservation. Science-based information is an important pillar for motivation towards preservation, since if ecological, economic and social values are ranked, governments and citizens will have clearer technical elements for better decision making.

Given that anthropic pressures generate great changes in ecosystems, it is necessary to analyze the components of the planning and management of the coastal zone arising from the social environment [55]. The Decalogue for the Integrated Management of Coastal Areas, proposed by Barragán [56], is a tool that links public policy processes, based on ten structural elements of the legal-administrative subsystem of a zone, allowing decision

makers to assess and compare the spaces in relation to their objectives. The ten elements of this decalogue are as follows: policy (explicit government policies directed to the management of coastal areas), participation (institutional and social support for public policies), normative (specific laws for coastal management), institutions (part of the public administration concerned with coastal marine spaces or resources), managers (government and stakeholders), information (physical, social-economic and administrative knowledge of the area), resources (economic tools that help apply and develop a management model), education (initiatives or proposals that promote education towards sustainability), strategies (for coastal management), instruments (tools such as zoning, concessions and authorizations for the use of coastal resources). In La Mancha, this decalogue includes the following elements:

- Policy: National Policy for Seas and Coasts of Mexico-Proposal of the Intersecretarial Commission for Sustainable Management of Seas and Coasts-2018 [57]; National Development Plan, 2019–2024 [58]; National Forestry Program 2020–2024 [59].
- Participation: representatives of the local, state and federal government; federation of fishermen; ecotourism organizations; academic institutions; grouping of fishing cooperatives with political interests; civil society [60].
- Normative: General Law of Ecological Balance and Environmental Protection (LEG-EEPA); National Water Law; Federal Law of the Sea; Sustainable Rural Development Law; Sustainable Forestry Development Law; General Wildlife Law; Fishing Law; Federal Law of Rights; General Law of Human Settlements; Mexican Official Standard NOM-059-SEMARNAT-200 (Environmental Protection-Mexican native species of wild flora and fauna); NOM-126-SEMARNAT-2000; Official Mexican Standard NOM-022-SEMARNAT-2003; Official Mexican Standard NOM-001-SEMARNAT-1996; Mexican Official Standard NOM-075-SEMARNAT-1994 [61].
- Institutions: Secretary of the Environment and Natural Resources (SEMARNAT); Secretariat of Agriculture, Livestock, Rural Development, Fisheries and Food (CONAPESCA) [62]; marine secretary; Ministry of Communications and Transportation; Secretary of Tourism; Secretariat of Health; Office of the Attorney General of the Republic; Secretariat of Management for Environmental Protection; General Directorate of the Federal Maritime Terrestrial Zone and Coastal Environments; National Water Commission; National Commission for Protected Natural Areas; Federal Attorney for Environmental Protection; National Institute of Ecology [61].
- Managers: administration of the Institute of the Ecology Sector of the Institute of Ecology; Secretariat of the Environment and Natural Resources (SEMARNAT); ZOFE-MATAC Federal Maritime Terrestrial Zone, dependency of SEMARNAT; Municipality of Actopan; State Coordination of the Environment (SEDERE) [60]; *La Mancha en Movimiento S.S.S.* [63]; PROFEPA Social Participation Committee; state nuclear power plant *Nucleoeléctrica Laguna Verde* (Federal Electricity Commission); Directorate of Fisheries State Agency; National Water Commission [61].
- Information: Community and Sustainable Development Program and Management Plan for the Protection and Conservation of the La Mancha-El Llano Ramsar Site [64]; La Mancha-El Llano Community Management Plan. In Search of a Sustainable Coastal Development [51]; Strategy for Comprehensive Coastal Management-The Municipal Approach [60]; Environments of Veracruz. The coast of La Mancha [62]; government; general bibliography.
- Resources: no specific data.
- Education: CICOLMA (La Mancha Coastal Research Center); Group La Mancha [61]; INECOL [51]; *La Mancha en Movimiento S.S.S.* [63]; Institute of Anthropology; Universidad Veracruzana [61].
- Strategies: conservation and management program for the blue crab; productive enclosures in rivers, lagoons and ponds; sowing and propagation of native species in nurseries [64]; ecotourism [13]; Strategy for Comprehensive Coastal Management-The Municipal Approach [64].

- Instruments: La Mancha-El Llano Community Management Plan [51].

*3.6. Summary of the DESCR Framework*

Being a coastal area, marine physical parameters such as wind, waves, tides and storm surge are the main processes regulating the lagoon systems. These natural drivers, as well as being pressures, are inherent to the system. Human activities in La Mancha are anthropic drivers: the urban project of recent years is a potential risk due to the alterations to the natural conditions of the ecosystems and changes in land use, among others. Exchanges refer to the interaction between the drivers and the system, where in La Mancha, the lagoon's own biophysical processes regulate hydrodynamics, sea level rise and the opening and closing of the inlet of the lagoon. Energetic periods of winds and waves cause the lagoon inlet to close, due to intense sediment transport, and the water level in the lagoon increases, favoring the development of mangrove species. The level of flooding and the fluctuations in salinity are factors that condition the growth of some species of fish.

The state of the environment in the lagoon area has been modified by anthropic drivers: the opening of the lagoon inlet by the fisherman alters its natural hydrological regime, and the urban project produces an imbalance in the natural dynamics of the system, which could cause conflict in the future if adequate management of the project is not afforded.

Responses should include strengthening local education programs in the area. Such actions will help to temper the lack of socio-environmental awareness and respect for laws. Continuous environmental monitoring will produce reliable information that should be made public, reduce uncertainties and ensure that appropriate action is taken to adapt measures where necessary. Efforts should be made to reduce socio-economic inequality. By encouraging other economic activities, pressure on coastal ecosystems will decrease, and economic vulnerability will be reduced due to the emergent urbanization projects. Measures should be put in place for wastewater treatment in the basin to control its consequences. Mitigation and management plans should be implemented to address pressure in emerging problems.

The following diagram (Figure 12) summarizes the DESCR framework for La Mancha.

From the elements listed in Figure 12, the interaction of some natural and anthropic drivers, through energy and matter exchanges in La Mancha, may induce coastal squeeze by the following:

- Alteration in the natural patterns of the physicochemical parameters (e.g., salinity and pH) due to the forced opening of the inlet;
- Sea level rise;
- Sediment deficit;
- Sediment availability reduction;
- Changes in native species coverage;
- Land use changes, including development of urbanization.

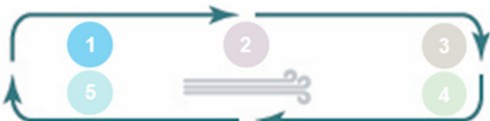

**Figure 12.** DESCR for La Mancha, Veracruz.

### 4. Conclusions

The DESCR framework is a useful tool to evaluate processes in this study area, taking into account the causes that trigger variations in the state of the environment, the physical environment with its natural dynamism and the social actors who will be responsible for formulating solutions to mitigate the problems detected.

In La Mancha, the natural dynamics of the ecosystems have been altered, by natural and anthropic factors, such as the artificial opening of the inlet. It is necessary to determine the long-term consequences of these actions in order to provide adequate elements for decision making. On the other hand, the urban development plans put the environmental integrity of La Mancha at risk. A lack of commitment to protecting the ecological resources and environmental services provided by the ecosystems of La Mancha is evident.

Although further scientific-based quantitative analysis of the interactions and responses is needed, the conceptual implementation of the DESCR framework allows a diagnosis to be made, and the identification of key interactions that occur in the system. The results presented here should serve as a basis for implementing better coastal management strategies. Projects with evaluation tools such as the DESCR framework can help link the local community with the scientific community and the authorities and provide means for developing future sustainable projects in coastal areas. Innovative forms of linking stakeholders, such as environmental education towards sustainability, seeking a high level of scope for these projects and monitoring over time will yield tangible results for a better use of the environmental resources of the La Mancha lagoon region.

**Author Contributions:** Conceptualization, R.S.; methodology, R.S., V.C.; formal analysis, S.C.A.; investigation, S.C.A.; resources, R.S.; writing—original draft preparation, S.C.A., V.C.; writing—review and editing, S.C.A., V.C., R.S., M.L.M., G.A.; supervision, V.C., R.S., M.L.M., G.A.; project administration, R.S.; funding acquisition, R.S., G.A. All authors have read and agreed to the published version of the manuscript.

**Funding:** CONACYT-SENER-Sustentabilidad Energética project: FSE-2014-06-249795 Centro Mexicano de Innovación en Energía del Océano (CEMIE-Océano).

**Acknowledgments:** This work is a contribution to the PAI Research Group RNM-328 (Andalusia, Spain).

**Conflicts of Interest:** The authors declare no conflict of interest.

## Appendix A

Numerous research works have been carried out in the area of La Mancha since the 1970s. Some of these studies analyzed its ecological and biological characteristics, determined by the presence of phytoplankton, vegetation, primary productivity, zooplankton, benthos, nekton, birds and pollution, as well as studies on fishing activity and aquaculture (e.g., [65–70]). The hydro-morphological dynamics and coastal management in the area have also been studied. Lists of relevant comprehensive studies of the local dynamics (articles and theses) are presented in Tables A1 and A2, respectively.

**Table A1.** List of relevant articles of La Mancha.

| Author, Year | Title | Main Contributions |
|---|---|---|
| Moreno-Casasola et al., 2006 [51] | Plan de manejo comunitario La Mancha-El Llano. En busca de un desarrollo costero sustentable. Estrategias para el manejo integral de la zona costera: un enfoque municipal | A management strategy based on the environmental protection and the experiences of the Latin American Forum of Environmental Sciences. |
| Moreno-Casasola, 2006 [62] | Entornos Veracruzanos. La costa de La Mancha | This book describes the main characteristics, physical, ecological, social and cultural, in great detail. |
| Moreno-Casasola et al., 2007 [52] | Los conflictos de la conservación: el caso de La Mancha. Hacia una cultura de conservación de la diversidad biológica | A view of the experiences of the social sectors of La Mancha and the conflicts of interests between them and the government regarding the introduction of the La Mancha-El Llano Community Management Plan. |
| Moreno-Casasola and Salinas Pulido, 2007 [64] | Programa de desarrollo comunitario sustentable y plan de manejo para la protección y conservación del Sitio Ramsar La Mancha-El Llano | An environmental conservation and community project for La Mancha, Veracruz, based on the sustainable management models of three groups of people from the local community. |
| Hesp and Martínez, 2008 [16] | Transverse dune trailing ridges and vegetation succession | A description of features, such as evolution and vegetation, of the Farallon dunes located in La Mancha, Veracruz. |
| Utrera-López and Moreno-Casasola, 2008 [20] | Mangrove litter dynamics in La Mancha Lagoon, Veracruz, Mexico | A description of litter dynamics among mangrove types in La Mancha, to help understand functional heterogeneity within this coastal ecosystem. |
| Psuty et al., 2009 [15] | Interaction of alongshore sediment transport and habitat conditions at Laguna La Mancha, Veracruz, Mexico | An overview of the hydrological regime, the dynamic of sediments and the ecological features in La Mancha. |
| Mata et al., 2011 [71] | Floristic composition and soil characteristics of tropical freshwater forested wetlands of Veracruz on the coastal plain of the Gulf of Mexico | Analysis of the geomorphological setting, influence and soil properties on the structure of vegetation of five coastal lagoons in Veracruz, one being La Mancha. |

**Table A1.** *Cont.*

| Author, Year | Title | Main Contributions |
|---|---|---|
| Martínez et al., 2014 [9] | Land use changes and sea level rise may induce a "coastal squeeze" on the coasts of Veracruz, Mexico | Analysis of the coastal line geodynamics and geodynamic trends to model niches under SLR scenarios. |
| Ruiz and López-Portillo, 2014 [72] | Variación espacio-temporal de la comunidad de macroinvertebrados epibiontes en las raíces del mangle rojo Rhizophora mangle (Rhizophoraceae) en la laguna costera de La Mancha, Veracruz, México | An analysis of spatiotemporal variations of epibiont macroinvertebrates in red mangrove roots (Rhizophoraceae), based on the hydrological dynamics of the system. |
| Ramírez Méndez et al., 2015 [73] | Estudio de la dinámica y calidad de agua en la laguna de La Mancha, Veracruz | A study of the dynamics of hydrodynamic, morphodynamic and hydrological conditions and physicochemical parameters of La Mancha and its littoral cell. |
| Chávez et al., 2017 [12] | Impact of Inlet Management on the Resilience of a Coastal Lagoon: La Mancha, Veracruz, Mexico | A study on features of La Mancha lagoon, such as ecosystem vulnerability, physical processes, such as erosion and accretion of the beach, inlet dynamics, and hydrodynamics of circulation patterns in the lagoon. |
| Rivera et al., 2019 [36] | Modelling the effects of the artificial opening of an inlet: Salinity distribution in a coastal lagoon | Numerical results are used to analyze and describe changes in the salinity distribution in La Mancha lagoon, showing results which would be useful in developing an adequate management plan. |
| Gómez Ramírez, 2020 [53] | El estudio de los ciclones tropicales se minimizó, en la manifestación del impacto ambiental para el Proyecto Diada La Mancha, en la costa del municipio de Actopan, Estado de Veracruz | A review of the problems caused by the lack of a serious environmental impact study prior to the development of the La Mancha-Diada project. |

**Table A2.** List of relevant theses of La Mancha.

| Author, Year | Title | Main Contributions |
|---|---|---|
| Cortés López, 2017 [44] | Desarrollo de un Índice de Riesgo sobre la ocurrencia de Opresión Costera en el centro-norte del Estado de Veracruz Master Thesis | Development of a risk index, based on the occurrence of coastal oppression, considering the evolution of the coastline and of the sediments in 14 beaches, one being La Mancha. |
| Chávez, 2018 [46] | Balance Hidrodinámico en Humedales Costeros y su valor como elemento de protección litoral PhD Thesis | Analysis and characterization of the physical protection against floods provided by wetlands, based on the monitoring of the main physical processes and the determination of their balances. |
| Ramírez Méndez, 2018 [18] | Rumbo a un Plan de Manejo Integral de la laguna de La Mancha, Veracruz Master Thesis | Study of physical, ecological and social parameters of La Mancha, Veracruz, in order to extend a management plan for this zone. |

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
