# Peer review of "Understanding the Dynamics of a Coastal Lagoon: Drivers, Exchanges, State of the Environment, Consequences and Responses"

_geosciences, doi:10.3390/geosciences11080301_

Round 1

Reviewer 1 Report

Line by Line comments

General comments: The policy aspects of this are generally strong, however the review of the geology and site conditions remain weak, particularly the wording around the inlet opening and closing.

Two style points permeate this paper. I have marked some of these up in the PDF provided (forgive the onscreen stylus pen handwriting!)

  1. the use of short paragraphs. In many case these need to be combined into longer paragraphs
  2. The use of hanging phrases and prepositions at the start of sentences. These sentences should be reordered or reworded in most cases (or many of the phrases can be cut)

Line 69 – See comment on PDF, this is often wording geologically. A better explanation of the processes and morphology is needed.

Line 80 – Good example of the hanging phrase today. Reorder to ‘Tourisim in the vicinity of La Mancha is managed….’

Line 94 – Need more detail on the lagoon. Size of the lagoon? Depths? Tidal range when open?

Line 158 – List the driver first and then explain each

Line 186 – Awkward paragraph. Also, use a different word on line 187 (substantial?) instead of significant

Line 195 – Why I sea level rise here lower than eustatic sea level rise? Tectonics? Uplift?

Line 203 – what natural regimes are modified?

Line 215 – use human less in that sentence

Line 227/229 – another good example to reorder- move ‘In 2018’ later in the sentence

Line 258 – combine those paragraphs

Line 266 – what is the elevation relative to? 120cm NAVD88? MLLW? MHHW?

269 – See the comments on PDF, needs reordering

Line 297-298 – How do you know?

Line 305 – can you provide figure here? Profile of the beach? Map of the inlet area?

Line 314 – reorder

Figure 9 – Scale Bar would be helpful

348 – extending alongshore? Offshore (i.e. prograding)

352 – remove drift – replace with longshore transport

354 – this paragraph needs a figure. The wording here on ‘bare sand’ is very very awkward

400 – Reorder: ‘Human activity has played an important role in the destabilization of the beach and dunes, creating…. (finish the sentence)’

Author Response

Line by Line comments

General comments: The policy aspects of this are generally strong, however the review of the geology and site conditions remain weak, particularly the wording around the inlet opening and closing.

Two style points permeate this paper. I have marked some of these up in the PDF provided (forgive the onscreen stylus pen handwriting!)

  1. the use of short paragraphs. In many case these need to be combined into longer paragraphs
  2. The use of hanging phrases and prepositions at the start of sentences. These sentences should be reordered or reworded in most cases (or many of the phrases can be cut)

Thank you for your comments, the changes made to address your suggestion were highlighted in yellow. Some editorial revisions were made (text in blue).

Line 69 – See comment on PDF, this is often wording geologically. A better explanation of the processes and morphology is needed. This paragraph and section 2.1 Study area were reordered, to present the details of the lagoon in section 2.1. (lines 80-92)

Line 80 – Good example of the hanging phrase today. Reorder to ‘Tourism in the vicinity of La Mancha is managed….’ Done line: 62                                                                  

Line 94 – Need more detail on the lagoon. Size of the lagoon? Depths? Tidal range when open? The introduction and this section 2.1 Study area were reordered, to present the details of the lagoon in section 2.1. (lines 80-92)

Line 158 – List the driver first and then explain each Done line:160-162

Line 186 – Awkward paragraph. Also, use a different word on line 187 (substantial?) instead of significant Done lines:190-193

Line 195 – Why I sea level rise here lower than eustatic sea level rise? Tectonics? Uplift? Done, this edited paragraph was moved to lines: 258-267

Line 203 – what natural regimes are modified? The sentence was reworded Done: line 200-201

Line 215 – use human less in that sentence The sentence was reworded Done: line 216-218

Line 227/229 – another good example to reorder- move ‘In 2018’ later in the sentence The sentence was reworded. Done: lines 228-231

Line 258 – combine those paragraphs. Done: lines 269-275

Line 266 – what is the elevation relative to? 120cm NAVD88? MLLW? MHHW? Done: lines 277-279

269 – See the comments on PDF, needs reordering. These sentences were reorganized. Lines 280-283

Line 297-298 – How do you know? Done line: This statement was deleted.

Line 305 – can you provide figure here? Profile of the beach? Map of the inlet area? Done, Figure 8 was added.

Figure 9 – Scale Bar would be helpful Done

348 – extending alongshore? Offshore (i.e. prograding). Done line: 351

354 – this paragraph needs a figure. The wording here on ‘bare sand’ is very very awkward. The paragraph was edited. No data was available for the authors to present a figure, but the reference where figures of these findings are presented was clearly cited. Lines: 355-357.

400 – Reorder: ‘Human activity has played an important role in the destabilization of the beach and dunes, creating…. (finish the sentence)’ The text was reworded. Done lines: 394-397

PDF COMMENTS

Line 160 - Era5 definition Done line:165-168

Line 201 such as…Done. Lines 198-199

Reviewer 2 Report

The simplest way to describe the shortcomings of the paper is to say that it lacks robustness.

For every aspect of the La Mancha system covered in the analysis there are so many variables that may be important and yet are not mentioned, much less discussed in detail. The DPSIR framework is primarily intended as a tool to facilitate decision-making across sectoral boundaries, between disciplines and with varied stakeholder. And as such needs to distil complex physical and anthropogenic interactions into digestible forms for shared understanding and planning. As presented in the this paper, the framework (its DESCR version here) is a superficial and descriptive treatment of the issues. nearly every section has some references to relevant literature being held up as evidence for problems or impacts at La Mancha.

The concept for the paper is sound, but the execution is not strong enough. An argument may be made that the evidence for the specific site and for all the relevant variables, does not exist. That does not mean they cannot be discussed in a conceptual way, building a conceptual model of the system and subsystems for La Mancha that would be very useful for identifying key pieces of research that should be prioritised, while also providing a robust basis on which to develop more convincing DESCR.

Author Response

The simplest way to describe the shortcomings of the paper is to say that it lacks robustness.

For every aspect of the La Mancha system covered in the analysis there are so many variables that may be important and yet are not mentioned, much less discussed in detail. The DPSIR framework is primarily intended as a tool to facilitate decision-making across sectoral boundaries, between disciplines and with varied stakeholder. And as such needs to distil complex physical and anthropogenic interactions into digestible forms for shared understanding and planning. As presented in the this paper, the framework (its DESCR version here) is a superficial and descriptive treatment of the issues. nearly every section has some references to relevant literature being held up as evidence for problems or impacts at La Mancha.

The concept for the paper is sound, but the execution is not strong enough. An argument may be made that the evidence for the specific site and for all the relevant variables, does not exist. That does not mean they cannot be discussed in a conceptual way, building a conceptual model of the system and subsystems for La Mancha that would be very useful for identifying key pieces of research that should be prioritised, while also providing a robust basis on which to develop more convincing DESCR.

Thank you for your observations. The changes made to address your suggestion were highlighted in green. Some editorial revisions were made (text in blue).

More details were added along this review article, also including reviewer 1 comments. A discussion on the summary of the results was added to improve the manuscript.

Reviewer 3 Report

Overall, I think the topic of the manuscript is interesting, very valuable, and is potentially helping to provide strategies based on the integration of different sources of information about Understanding the Dynamics of a Coastal Lagoon using the DESCR tool. Nevertheless, the manuscript needs some clarifications and adjustments. I have focused on the methods used in the review.

  1. The authors mentioned DESCR Framework was used, but they did not provide sufficient details regarding how the DESCR tool was used.
  • Which specifical journals in Scopus and Google Scholar databases were consulted?
  • How were many theses of master’s and Ph.D. consulted? Please, quotes the most relevant in the references. List these references in the first paragraph of section 2.2 entitled DESCR Framework (page 5). Perhaps, contributions of authors reviewed can be mentioned in a table as supplementary material.
  • Detail of natural and anthropogenic threats selected are not described. Why threats, such as flooding, erosion, mass removal, or sea-level rise are not analyzed? Please, mention it.
  1. A specific annex or supplementary material regarding coastal squeeze could be a relevant analysis in this study area. The authors should offer some specific details regarding what kind of features are present in La Mancha lagoon, Veracruz.
  2. The authors must mention in a detailed way the source of primary or secondary data assessment used. Trustworthiness is one of the main criteria for qualitative research. The authors should revise the section of Materials and Methods to improve the trustworthiness of their manuscript.

Author Response

Dear Reviewer

thank you for your observations, we answered to all of them...please see attached file and manuscript.

Best regards.

Round 2

Reviewer 2 Report

Dear Authors,

I did not provide detailed comments in the original feedback, because I indicated that 'major revisions' were necessary (in my opinion). The revised manuscript is not a major revision on the original, so my recommendation and comments still stand. It might be that my comments were not clear enough: I have added a few lines of clarification below (in bold).

For every aspect of the La Mancha system covered in the analysis there are so many variables that may be important and yet are not mentioned, much less discussed in detail. The DPSIR framework is primarily intended as a tool to facilitate decision-making across sectoral boundaries, between disciplines and with varied stakeholder. And as such needs to distil complex physical and anthropogenic interactions into digestible forms for shared understanding and planning. As presented in the this paper, the framework (its DESCR version here) is a superficial and descriptive treatment of the issues. This is a major flaw. It is worst than poor science, it is counter-productive, in that it undermines the position of science as a basis for evidence-based decision-making. 

Nearly every section has some references to relevant literature being held up as evidence for problems or impacts at La Mancha. As such it is mis-leading, there needs to be much clearer differentiation between what is evidence for the site and what are statements about scientific understanding of the processes, from the literature.

The concept for the paper is sound, but the execution is not strong enough. That is to say the paper is not good enough, and requires a major rethink and revision. This is something I would expect to take weeks (if not months), certainly not a few days.

An argument may be made that the evidence for the specific site, and for all the relevant variables, does not exist. That does not mean they cannot be discussed in a conceptual way, building a conceptual model of the system and subsystems for La Mancha that would be very useful for identifying key pieces of research that should be prioritised, while also providing a robust basis on which to develop a more convincing DESCR. That is to say, the authors need to present a much more robust conceptual model of the system and subsystems for La Mancha, before presenting their final figure as a summary of that model. When I say 'robust', I mean including a detailed discussion of site-specific variables and drivers AND integrating general scientific understanding from the literature where site-specific data is not available.

In essence, from the original manuscript very little would remain (only the central concept for the paper), leading to a completely new paper being developed. This, in my opinion, is what is required to publish this work and have it be impactful in a meaningful way. 

Author Response

Thank you for your clarifications.

The aim of this review type paper is to present the state of the art. Therefore, after the conceptual implementation of the DESCR model, further scientific-based quantitative analysis of the interactions and responses is needed.

We believe that this is a very useful review tool to make a diagnosis and to identify the key interactions that occur in a system, and to propose a general list of possible responses to improve the health of a system and/or to prevent further undesired impacts.

Given that a process of “major revision” for this journal is expected to take 10 days, we made substantial changes marked in green, with the aim of improving the manuscript line with your observations, rather than write “a completely new paper”.